# Manipulation of Medicinal Products for Oral Administration to Paediatric Patients at a German University Hospital: An Observational Study

**DOI:** 10.3390/pharmaceutics12060583

**Published:** 2020-06-23

**Authors:** Julia Zahn, André Hoerning, Regina Trollmann, Wolfgang Rascher, Antje Neubert

**Affiliations:** Department of Paediatrics and Adolescent Medicine, Universitätsklinikum Erlangen, 91054 Erlangen, Germany; andre.hoerning@uk-erlangen.de (A.H.); regina.trollmann@uk-erlangen.de (R.T.); wolfgang.rascher@uk-erlangen.de (W.R.)

**Keywords:** off-label use/manipulation, paediatrics, age-appropriate formulation, oral drug administration, hospital, drug manipulation

## Abstract

Pharmacotherapy in children requires medicinal products in age-appropriate dosage forms and flexible dose strengths. Healthcare professionals often encounter a lack of licensed and commercially available formulations, which results in the need for manipulation. This study aimed to investigate the nature, frequency and preventability of the manipulation of medicinal products before oral drug administration to paediatric inpatients in Germany. A prospective, direct observational approach was used. Two thousand and three medication preparation processes (MPP) in 193 patients were included in the analysis. Medicines were manipulated in 37% of oral administrations, affecting 57% of the patients. The percentage of manipulations was highest in infants/toddlers (42%) and lowest in adolescents (31%). Antiepileptics were most frequently manipulated (27%), followed by vitamins (20%) and drugs for acid-related disorders (13%). Fifty-six per cent of all manipulations were off-label. In 71% of these, no alternative appropriate medicinal product was commercially available. These results demonstrate that the manipulation of medicinal products before oral administration is common in paediatric wards in Germany. About half of the manipulations were off-label, indicating that no suitable formulation was available. Evidence-based guidelines for manipulations are required, with the overall aim of improving the safety of paediatric drug therapy.

## 1. Introduction

Adequate and safe pharmacotherapy in children requires evidence-based dosing guidelines as well as licensed medicinal products in age-appropriate dosage forms and flexible dose strengths [1,2,3]. The paediatric population is characterised by continuous growth and developmental changes [4]. Hence, pharmacotherapy needs to be individually tailored to the patient to improve adherence and therapeutic outcomes and to reduce the risk of medication errors [5]. In addition to the correct dose, the acceptability of the dosage form and taste, and the individual’s ability to handle a medicine are important points to consider when prescribing a drug to a child. Young children and patients with special conditions, e.g., the presence of a feeding tube, are not able to take whole tablets or capsules [3,6,7]. However, healthcare professionals and caregivers often face a lack of commercially available, age-appropriate formulations in the required dose strength. Consequently, the manipulation of medicinal products to extract a part of the whole dosage form, to facilitate administration or to improve acceptability is common practice in paediatric drug therapy [8,9,10,11,12,13].

The term manipulation includes, among other things, splitting, crushing or suspending tablets, and opening capsules and suspending the content in liquids. These steps are rarely supported by the manufacturer’s summary of product characteristics (SmPC) or other guidelines [8,9,14]. It has previously been reported that manipulations of medicines increase the risk of adverse events (AE) [12]. They can affect dosing accuracy, bioavailability and integrity of the dosage form to an unknown extent. As a result, sub-therapeutic or toxic doses may be administered to vulnerable patient populations [12,15,16,17]. A randomised study in patients aged from 10–16 years showed that administration of crushed lopinavir/ritonavir tablets significantly reduced drug exposure compared to patients receiving whole tablets [18].

Several publications revealed that dose accuracy and weight uniformity are highly dependent on the manipulation method used [19,20,21]. Thus, even if a manipulation is supported by the SmPC, it poses an additional risk to the patient. Furthermore, bioavailability and pharmacokinetic data are lacking for manipulated dosage forms [14,19]. 

A study from Norway, which was conducted for eight weeks in two hospitals, showed that 17% of 3070 oral drug administrations to paediatric patients were not carried out as described in the relevant SmPC [9]. Recently, van der Vossen et al. [22] revealed that in six weeks of observation, 37% of oral administrations to paediatric inpatients in the Netherlands were manipulated. However, the data does not allow for a general statement that can be applied to other countries. Moreover, most studies were not designed to determine the actual frequency of manipulation, as their study protocol did not include observation of all manipulations within the observation period. The aim of this study was, therefore, to investigate the nature, frequency and prevalence of the manipulation of medicinal products prior to oral administration in paediatric wards at a German university hospital by direct observation, and to determine the preventability of these manipulations. 

## 2. Methods

### 2.1. Design

A prospective, cross-sectional, monocentric observational study was conducted in two general paediatric wards at the Department of Paediatric and Adolescent Medicine of the University Hospital in Erlangen, Germany. One ward focuses on infectious and gastroenterological diseases (23 beds, Ward A) and the other ward specialises in neuropaediatrics and metabolic diseases (23 beds, Ward B). 

### 2.2. Inclusion and Exclusion Criteria

All patients aged less than 18 years receiving oral medication on the day of observation were included. All medication preparation processes (MPP) observed in those patients were documented. MPP for routes of administration other than oral medication as well as nutritional supplements were excluded. 

### 2.3. Data Collection

On randomly assigned days between January 2019 and May 2019, a trained pharmacist attended the MPP by the nursing staff in the pharmacy room of the respective ward. Random selection of the days of observation was used to detect the MPP of as many different patients as possible within the study period. On observational days, the observer documented the MPP of all patients on the ward receiving oral medication. Manipulations were identified prospectively via direct observation of the nursing staff while they dispensed the medication. Data from the observations were documented on a study-specific case report form.

### 2.4. Ethical Approval

The Ethics Committee of the Friedrich-Alexander-University approved the study protocol (Application nr. 114_18 B, 28 March 2018). 

Nurses were informed about the scope of the study and gave verbal consent before being monitored for the study.

The protocol specified that the observer was obliged to intervene in the event of a potential risk of the MPP to the patient or the nurse. This meant that the nursing staff or physician had to be informed and the error would be avoided before it affected the patient. An example of a harmful situation is when the dispensed dose is not in line with the prescribed dose. 

### 2.5. Definitions

MPP was defined as all processes related to the preparation and dispensing of the patient’s individually prescribed dose by the nursing staff prior to oral drug administration. 

Manipulation was defined as “all activities prior to administration that are undertaken in order to administer the medicine to the patient using an alternative strategy (e.g., in order to improve patient acceptability or adjust the dose)”. This definition is derived from the guidelines on the pharmaceutical development of medicines for paediatric use by the Committee on Medicinal Products for Human Use (CHMP) and the Paediatric Committee (PDCO) [23]. Within the scope of this study, this comprises all manipulations, both for extracting a proportion of the whole dosage form, as well as modifying the dosage form to facilitate administration, e.g., via a feeding tube. Examples of manipulations of oral dosage forms are shown in Table 1. 

One manipulation could comprise different steps and could, therefore, be a combination of different manipulation types. If this was the case, the first step was considered as the main manipulation step and the manipulation was classified accordingly.

In this study, manipulation is considered “off-label” if at least one of the activities carried out within the manipulation process is not indicated in the relevant SmPC. For example, in the case where a tablet is split although the SmPC does not contain any information regarding splitting of the tablet. 

A manipulation was considered “preventable” if an alternative medicinal product that did not need to be manipulated was commercially available in the German market. The availability of extemporaneous formulations was not taken into account.

### 2.6. Data Analysis

Duplicate manipulations (i.e., identical patient, hospital stay, medication and dosage) were excluded from the data analysis to avoid biases by patients with an extended length of hospital stay or polypharmacy. 

The reason for carrying out a manipulation was determined and classified as follows. If only a part of the whole dosage form was administered, the reason for this manipulation was considered as “inappropriate strength”. If the dosage form was administered entirely but was manipulated prior to administration, e.g., it was crushed or dispersed because the patient experienced swallowing difficulties or was fed via a tube, “inappropriate dosage form” was chosen as the reason. If both reasons for manipulation applied, inappropriate dosage form was considered as the main reason.

The prevalence and frequency of the manipulations were calculated and stratified by age group, type of manipulation and number of manipulation steps. Prevalence of manipulations was defined as the quotient of patients receiving at least one manipulated oral medication and patients receiving no manipulated oral medication during the observational period (patients with manipulation/patients overall × 100%). 

For statistical analysis, IBM^®^ SPSS^®^ Statistics Version 24 was used. Differences between the groups of patients with and without manipulation were calculated using the Mann-Whitney U test for independent samples. Age group comparisons in terms of frequency of manipulations, the number of manipulation steps and the proportion of the dosage form administered were calculated using the Kruskal–Wallis test for independent samples and subsequent post-hoc testing (Bonferroni). The correlation between the age of the patients and the number of manipulation steps was done using Pearson’s bivariate regression. The significance level was set at α = 0.05 (two-tailed) for all tests.

## 3. Results

### 3.1. Prevalence of Manipulations

During the five-month observation period, a total of 2003 MPP in 193 patients were witnessed. After elimination of the duplicate manipulations, a total of 640 MPP in 193 patients (mean age: 6.8 years, min: 21 days, max: 17.9 years) were included in the analysis (Table 2). The prevalence of patients affected by at least one manipulation was 57.0% (110/193) overall and 25.4% (49/193) when only off-label manipulations were considered. Patients with manipulated medication were significantly younger, had less body weight, and their duration of stay was longer compared to patients without manipulated medication (*p* < 0.05, Table 2).

### 3.2. Frequency of Manipulations

Thirty-seven per cent (237/640) of the observed MPP included the manipulation of medicinal products. In 46.0% (109/237) of the manipulations, the affected patient was fed via a tube.

The frequency of manipulation was highest for infants and toddlers with 42.4%, followed by school children (6–11 years) with 38.7% and pre-school children (2–5 years) with 35.1%. Adolescents (12–17 years) were affected by manipulation in 30.9% of the oral administrations (*p* > 0.05, Table 2). 

### 3.3. Licensing Status and Preventability of Manipulations

In 20.8% (133/640) of the MPP, medicinal products were not manipulated according to what was described in the relevant SmPC. Thus, 56.1% of all manipulations observed were classified as off-label. Analysed by age group, pre-school children received off-label manipulations in 25.2%, school children in 22.6%, infants and toddlers in 19.5% and adolescents in 18.8% of MPP (Table 2). Of the administrations to patients with a feeding tube, 30.4% were manipulated off-label.

The flowchart in Figure 1 describes the distribution of manipulations classified by licensing status and preventability. Of the manipulations classified as off-label, 71.4% were non-preventable. Preventability of off-label manipulations increased by age group (Figure 2). A comprehensive list of the manipulated active substances classified by preventability and licensing status is provided in the Appendix A.

### 3.4. Active Substances and Therapeutic Subgroups Affected by Manipulations

Antiepileptics were most commonly manipulated (24.9%), followed by vitamins (19.8%) and drugs for acid-related disorders (12.7%) (Table 3). Cholecalciferol (n = 45), omeprazole (n = 27) and phenobarbital (n = 18) were the most frequently manipulated active substances. Manipulated medicinal products with modified drug release included Antra mups^®^ (omeprazole, n = 21), Orfiril^®^ long (valproic acid, n = 4), Beloc-zok^®^ (metoprolol-succinate, n = 1) and Circadin^®^ (melatonin, n = 1).

### 3.5. Dosage Forms Manipulated and Type of Manipulation

The most frequently manipulated dosage forms were tablets (173/237, 73.0%) and tablets/capsules/granules with modified drug release (34/237, 14.3%). Granules/tablets/powder for oral suspension were manipulated in 6.3% (15/237) of cases, and capsules in 1.3% (3/237) of cases.

The most frequently observed manipulation type was splitting of tablets (34%), followed by dispersing, suspending or dissolving of a solid dosage form in liquid (25%) (Figure 3).

In 18% of the cases, splitting alone did not provide the intended dose and a further manipulation step was used, e.g., suspension in liquid (14.8%) and withdrawing a defined volume of the suspension (3.0%). In 16% of the cases in which the dosage form was dispersed as a first step, another manipulation step for dose adjustment was performed. Other types of manipulation included dilution of a liquid dosage form in a larger volume (2.5%), crushing of tablets (0.8%) and subsequent manipulation (0.8%) or counting of minitablets (0.8%) and subsequent suspension of the defined number of minitablets in liquid (0.4%). The frequencies of all manipulation types are displayed in Table 4.

#### 3.5.1. Manipulation Type Classified by Age Group

In the infants/toddlers, school children and adolescents age groups, the most common manipulation type was splitting a tablet, as a one-step manipulation. In pre-school children, more than one-step manipulations (>1-step manipulation) were predominant. Figure 4 displays the main types of manipulations per age group classified by the number of manipulation steps. Manipulations in pre-school children included significantly more manipulation steps compared to adolescents (*p* < 0.05). The age of the patient correlated significantly with the number of manipulation steps (r = −0.160, *p* < 0.05). In general, the younger the patient, the more manipulation steps were observed.

#### 3.5.2. Proportion of the Dosage Form Administered

In 70.5% (167/237) of the manipulations, only part of the original dosage form was administered. Infants/toddlers and pre-school children received significantly smaller proportions of the original dosage forms compared to adolescents (*p* < 0.05, effect size (infants/toddlers vs. adolescents) = 0.27; effect size (children, pre-school vs. adolescents) = 0.33). In 83.1% (139/167) of the manipulations in which the original dosage form was only partly administered, the remaining proportion of the dosage form was discarded. 

### 3.6. Root Causes for Manipulation

The most frequent reason for carrying out manipulations in children below six years of age was the inappropriateness of the dosage form. In the age groups above six years of age, the available dose strength of the medicinal product was inappropriate more often than the dosage form itself (Figure 5).

## 4. Discussion

### 4.1. Frequency of Manipulations

To our knowledge, this is the first study to investigate the manipulation of oral medicinal products in paediatric wards by direct observation in Germany. Thirty-seven per cent of all orally dispensed medicinal products were manipulated before administration, and more than half of the patients received at least one manipulated medicine. 

These findings are in line with the results from a study by van der Vossen et al. [22] that used a comparable observational method. However, the number of observed oral administrations was much lower than in our study. Bjerknes et al. [9] found that 17% of 3070 observed oral administrations were not manipulated according to the SmPC, which is only slightly lower than the 20.8% of off-label manipulations found in our study. Van der Vossen et al. [22] reported that 14% (16/115) of observed oral administrations were not manipulated according to the SmPC. 

Classified by age group, the frequency of manipulation was highest in infants and toddlers (42.4%) and lowest in adolescents (30.9%). Surprisingly, school children showed a higher frequency of manipulation than pre-school children. This could be explained by the fact that young children are mostly accompanied by parents or caregivers, who may take over the administration process from the nurses. 

The frequency of off-label manipulations in this study is comparable to that found in a study by Bjerknes et al. [9], which showed similar numbers for infants and toddlers (23% vs. 20%). However, Bjerknes et al. reported a much lower frequency for adolescents, which could be due to differences in the patient cohort of the two studies.

### 4.2. Licensing Status and Preventability of Manipulations

About 44% of the manipulations observed were in line with the SmPC, meaning that the procedure used to obtain the appropriate dose was explicitly mentioned in the SmPC. 

According to the findings of van der Vossen et al. [22], one can expect that a correctly (according to the SmPC) performed manipulation does not affect therapeutic goals. For instance, it is considered acceptable to split cholecalciferol tablets containing 1000 IU instead of using commercially available tablets containing 500 IU cholecalciferol. The reason for this is mainly economically driven. One hundred tablets of the 1000 IU tablets cost 5.0 €, and 100 tablets of the 500 IU cost 4.62 €, which means that almost 50% of the cost is saved when the higher strength is being used. Even though the costs for cholecalciferol are relatively low, most infants and toddlers receive it as a prophylaxis of rickets, which leads to substantial savings over time if the higher strength is being used. The frequent use of cholecalciferol tablets can also be considered as a reason for the high proportion of one-step manipulations in infants and toddlers. Nevertheless, this study demonstrated that, overall, younger age is associated with an increased number of manipulation steps.

However, we need to keep in mind that each alteration to the original formulation constitutes an additional risk for error and introduces variability to the intended dose [20]. Therefore, careful judgement is needed, and it should be decided case by case whether preventable manipulations are being accepted in a hospital solely due to economic reasons. 

In order to avoid preventable manipulations, it is also important to provide the physicians and nurses with an overview of the available dosage forms and strengths appropriate for children. It has been shown that informational material regarding drug administration procedures on the wards is often outdated or not even available [17]. Therefore, integrating the information on drug dispensation and administration in paediatric wards as well as regular training should be part of each hospital quality management system. 

Fifty-six per cent of manipulations were not covered by the SmPC, or additional steps were taken before administration of the medicine to the patient. Only 29% (n = 38) of these would have been preventable by using an alternative commercially available medicinal product. Most of the non-preventable off-label manipulations were generic drugs. Among the preventable manipulations, three cases could have been avoided if a medicinal product licensed through a Paediatric Use Marketing Authorization (PUMA) had been used (Slenyto^®^ (melatonin), Hemangiol^®^ (propranolol) and Alkindi^®^ (hydrocortisone)). Another at the time of this study non-preventable manipulation became preventable because a PUMA drug became available (Kigabeq^®^ (vigabatrin)). 

These results illustrate the urgent need for more age-appropriate formulations and the importance of the PUMA. Unfortunately, among the pillars of the paediatric regulation, the PUMA is the most critical and is often considered as a “failure” [26,27]. The PUMA needs to become more attractive to pharmaceutical companies, for whom it is not of economic interest to develop a paediatric formulation of a generic drug. 

### 4.3. Active Substances and Therapeutic Subgroups Affected by Manipulation

Antiepileptics were found to be the most commonly manipulated therapeutic subgroup. The studies of Bjerknes et al. [9] and Richey et al. [8] also found that antiepileptics are manipulated frequently. Possible reasons might be the already high off-label use in this therapeutic area and the slow dose titration, which is typical for antiepileptics. Kuchenbuch et al. [28] found that 33.5% of prescribed antiepileptic drugs to children in outpatient care generated manipulations. Of these, 40% were inadequate, which is in line with our study. This also explains why the neuropaediatric ward showed a higher prevalence of off-label manipulations compared to the other ward that focuses on infectious and gastroenterological diseases. Children with epilepsy have an increased risk of medication errors due to complex medications, polypharmacy and multiple medical conditions. Therapeutic drug monitoring is recommended for most antiepileptic drugs [29]. As manipulations can affect dosing accuracy to an unknown extent, this highlights the necessity of studies to investigate the performance and clinical outcomes of manipulations, especially for drugs with a narrow therapeutic index. In this evaluation, about one-third of the manipulations affected narrow therapeutic index drugs (NTIs), if the therapeutic subgroups of antiepileptics, immunosuppressants and antihypertensives were considered as NTIs.

This study found two off-label manipulations of generic tablets of acetylsalicylic acid (ASS). In both cases, the tablet was dispersed in water and afterwards, a fraction was withdrawn from the dispersion. In one of the cases, the tablet was halved prior to the dispersion. Brustugun et al. [19,20] investigated the accuracy and precision of doses of ASS obtained by using different manipulation methods. One of the findings was that splitting the tablet before dispersion resulted in high variability in the obtained dose [20]. Based on the information provided in these studies, assumptions can be made about the performance of a manipulation with the identical active substance. However, the dosage form itself, tablet properties, excipients, particle size distribution, as well as the manipulation method, influence the behaviour of a tablet during division and dispersion [19,20]. As the availability of medicinal products for children most likely will not improve rapidly, manipulations will still be common in paediatric pharmacotherapy. Therefore, this kind of study is needed to predict the outcome of manipulations.

### 4.4. Dosage Forms and Type of Manipulation

Consistent with the findings of Bjerknes et al. [9] and Richey et al. [8], tablets were the most frequently manipulated dosage form, and the two main manipulation types were splitting of tablets and the dispersal of a solid dosage form in liquid. Surprisingly, dispersing tablets in liquid was more frequent in pre-school children than infants and toddlers, for whom splitting tablets was the most common manipulation. This finding may be explained by the fact mentioned above that children below one year of age receive 500 IU/day cholecalciferol for rickets prophylaxis, which is mostly provided by splitting a 1000 IU tablet in the hospital [30].

Manipulations are even more critical if they affect modified release formulations. In our study, this was the case for 14.3% of the manipulations and concerned active substances like omeprazole or melatonin. 

The proton pump inhibitor, omeprazole (Antra mups^®^ 10 mg/20 mg) was the second most frequently manipulated active substance. This tablet formulation is a Multiple Unit Pellets System (MUPS^®^) containing enteric-coated pellets that protect the acid-labile omeprazole against protonation in the stomach [31]. Boussery et al. [32] showed that a suspension formulation of omeprazole (extemporaneous formulation) seemed to be more favourable with regard to plasma concentration-time profiles compared to the MUPS^®^ formulation. This highlights the complexity of ensuring the efficacy of proton pump inhibitors when being manipulated. In this context, it is also vital to consider the mechanism of drug release in order to determine the potential impact of manipulation of a modified release formulation.

Another active substance with modified drug release manipulated in this study was Circadin^®^, a prolonged-release tablet formulation of melatonin. Chua et al. [33] demonstrated that the crushing of Circadin^®^ led to an immediate release of melatonin rather than the desired prolonged release. Since 2019, the PUMA product Slenyto^®^ (prolonged-release tablets of melatonin) has been available in the German market. However, although this medicinal product is licensed especially for paediatric patients with developmental disabilities, data for administration via tube is lacking. Therefore, administration via tube is considered off-label according to the manufacturer [34]. This case illustrates that health professionals are often left alone in their decision-making when drugs need to be administered via a tube. 

Furthermore, our study revealed that nearly half of the manipulations observed affected patients with a feeding tube. More than 60% of the manipulations affecting these patients were not covered by the respective SmPC. These numbers demonstrate the importance of drug-specific information on administration via tube provided by the manufacturer, which is particularly relevant for special patient groups, such as children or elderly patients. Therefore, we advocate that information on feasible manipulations should be made available in the SmPC. This information should address the divisibility of tablets, especially whether a tablet may be divided into equal doses, mortarability, and the possibility of dispersing a tablet in liquid. Furthermore, manufacturers are encouraged to address the option of administering a medicinal product via an enteral feeding tube and provide guidance, e.g., solvent or Charrière size of the tube.

### 4.5. Proportion of the Dosage Form Administered

In more than 50% of the manipulations, only a part of the original dosage form was administered to patients and the remaining parts were discarded. This can be explained by hospital guidelines that take into account hygienic and safety aspects and prevent the risk of mix-ups of different active substances. However, from an economic point of view, discarding high amounts of medicines leads to unnecessary costs for the hospital and health care system. Economic considerations should therefore be integrated when developing guidelines for the administration of medicines to children. If child-appropriate dosage forms in the required strengths were available, the high proportion of discard could be reduced.

### 4.6. Root Cause for Manipulation

The main reasons for manipulation before administration were the inappropriateness of the dosage form and the inappropriateness of the strength. This is in line with the findings of Bjerknes et al. [9]. As previously discussed by van der Vossen et al. [22], the presence of a feeding tube led to several manipulations of the dosage form. Nonetheless, it should be taken into account that other reasons for manipulations, e.g., taste masking or size adjustment may not have been documented due to the observational method used. 

In the younger age groups, the dosage form itself was the main reason for the manipulation, whereas for older children, the non-availability of the appropriate strengths was the leading cause. In addition, younger children had more manipulations comprising of more than one step. For instance, in the case of a 20-month old infant receiving 1.25 mg enalapril, the tablet was split first, and due to the non-ability to swallow, the tablet was dispersed in water as a second step.

This confirms that particularly younger children have specific needs in terms of the dosage form. It is well known that children below two years of age have difficulties in swallowing solid dosage forms (except minitablets) and that liquid formulations should not contain problematic excipients, e.g., benzoic acid or benzyl alcohol [13].

### 4.7. Strengths and Limitations

To our knowledge, this was the first study investigating the nature and frequency of manipulations in paediatric pharmacotherapy in Germany. Additionally, the preventability of manipulation by the use of an alternative medicinal product has been taken into account in suggesting strategies for improvement. Preventability by using extemporaneous formulation was not considered in this study.

Direct observation was used and the patient population in the wards that we observed was broad as they covered a wide range of paediatric specialties such as neuropaediatrics, gastroenterology, infectious and metabolic diseases. Furthermore, the study duration of five months was noticeably longer than other studies [8,9,22]. Thus, the results can be considered reliable and close to the reality for the described age groups. However, this study did not evaluate clinical outcomes following the administration of manipulated medicines, such as the occurrence of adverse events. 

The comparability of this study to other studies was limited because of differences in the definition of the manipulation used [8,9,22]. 

One limitation of the study was that the observation was only performed in two wards of a German university hospital. Furthermore, neither of the wards specialised in newborns, which led to a low number of observed MPP in this age group. Therefore, the prevalence of manipulations observed in newborns did not provide representative results. Additionally, the observations did not include administration by parents or caregivers. This suggests that the prevalence of manipulation may be even higher than seen in this study, especially in the younger age groups.

## 5. Conclusions

This study showed that manipulation of medicinal products prior to oral administration to paediatric inpatients is common practice in Germany. Half of the manipulations were not covered by the relevant SmPC (off-label). However, only 29% of these manipulations would have been prevented by using commercially available medicinal products, which indicates that there is still a significant lack of available paediatric formulations. The available alternative medicinal products include four of the six drugs that have been granted a PUMA to date. Although the prevention of some manipulations leads to higher costs, the safety of paediatric drug therapy should not be jeopardised by unnecessary manipulations. The development of age-appropriate formulations for generic drugs has to be further facilitated. In the meantime, evidence-based guidelines, as well as regular training of staff, are required for non-preventable manipulations. 

## Figures and Tables

**Figure 1 pharmaceutics-12-00583-f001:**
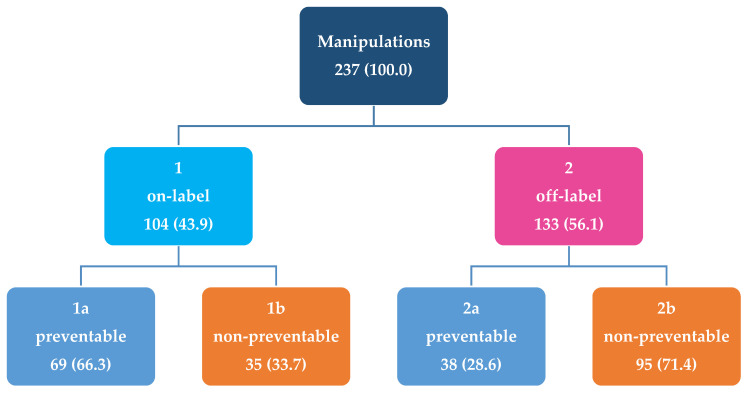
Flowchart of manipulations (n (%)) classified by licensing status (on- and off-label) and preventability (preventable and non-preventable).

**Figure 2 pharmaceutics-12-00583-f002:**
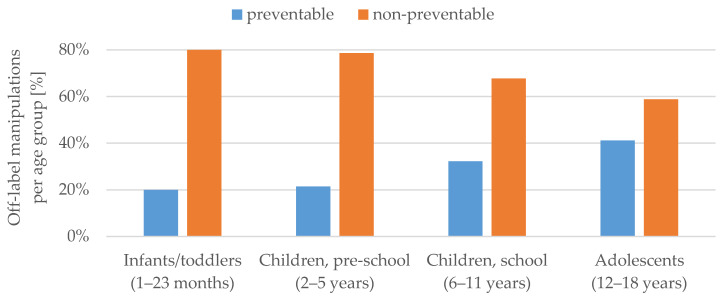
Off-label manipulations per age group (%) classified by preventability.

**Figure 3 pharmaceutics-12-00583-f003:**
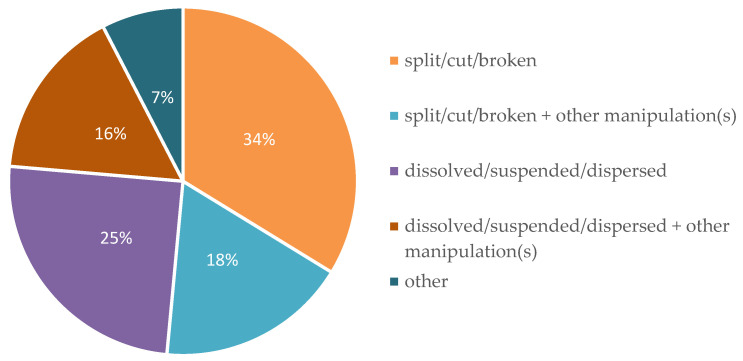
All manipulations observed classified by the type of manipulation.

**Figure 4 pharmaceutics-12-00583-f004:**
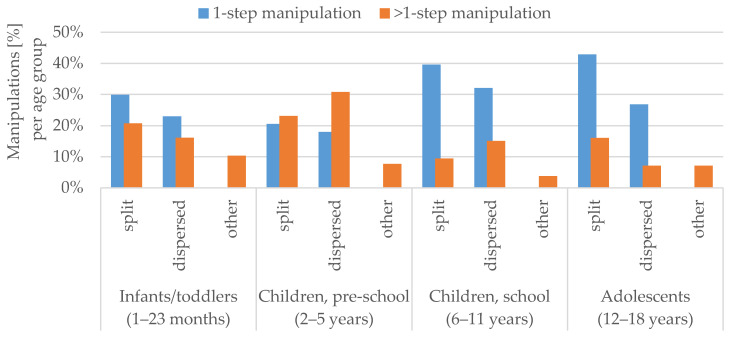
Main type of manipulation per age group (%) based on the total number of manipulations per age group classified by the number of manipulation steps (one or more than one-step).

**Figure 5 pharmaceutics-12-00583-f005:**
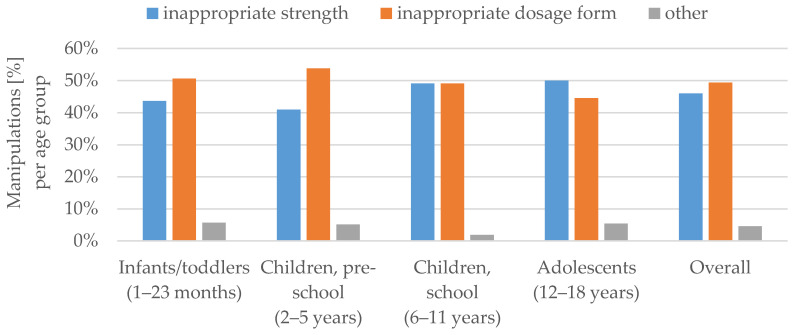
Root causes for manipulation (%) per age group.

**Table 1 pharmaceutics-12-00583-t001:** Examples of the manipulation of different dosage forms (adapted from [24]).

Dosage Form	Examples of Manipulations
Tablet	Splitting/cutting/crushing
	Dispersion in liquid
	Administration of a proportion/segment
	Dispersion in liquid and withdrawing a fraction
Capsule	Opening of a capsule
	Dispersion of content in liquid
	Administration of a proportion of the content
	Dispersion in liquid and withdrawing a fraction
Oral liquid	Dilution in a larger volume
	Administration of a fraction
Sachet with powder/granules	Dispersion of content in liquid and administration of a proportion
	Administration of a proportion of the content

**Table 2 pharmaceutics-12-00583-t002:** Descriptive analysis of patient characteristics, the frequency and prevalence of medication preparation processes (MPP) and manipulations (Man.) overall and by licensing status (off-label).

Specification	Patients Overall		Patients with Man.		Patients with Off-Label Man.		MPP		Man.		Off-Label Man.
	n (%)		n (%)		n (%)		n (%)		n (%)		n (%)
Overall											
Total	193 (100.0)		110 (57.0)		49 (25.4)		640 (100.0)		237 (37.0)		133 (20.8)
Ward											
A	88 (45.6)		52 (59.1)		16 (18.2)		307 (48.0)		114 (37.1)		57 (18.6)
B	105 (54.4)		58 (55.2)		33 (31.4)		333 (52.0)		123 (36.9)		76 (22.8)
Gender											
Female	88 (45.6)		48 (54.5)		21 (23.9)		312 (48.8)		106 (34.0)		59 (18.9)
Male	105 (54.4)		62 (59.0)		28 (26.7)		328 (51.3)		131 (39.9)		74 (22.6)
Age group											
Newborns (0–28 days)	2 (1.0)		2 (100.0)		0 (0.0)		6 (0.9)		2 (33.3)		0 (0.0)
Infants/toddlers (1–23 months)	58 (30.1)		42 (72.4)		11 (19.0)		205 (32.0)		87 (42.4)		40 (19.5)
Children, pre-school (2–5 years)	37 (19.2)		19 (52.4)		11 (29.7)		111 (17.3)		39 (35.1)		28 (25.2)
Children, school (6–11 years)	36 (18.7)		19 (52.8)		8 (22.2)		137 (21.4)		53 (38.7)		31 (22.6)
Adolescents (12–18 years)	60 (31.1)		28 (46.7)		19 (31.7)		181 (28.3)		56 (30.9)		34 (18.8)
Presence of feeding tube	34 (17.6)		31 (91.2)		15 (44.1)		230 (35.9)		109 (47.4)		70 (30.4)
	**Patients with Man.**					**Patients without Man.**					
	Mean	SD	Median	Min	Max	Mean	SD	Median	Min	Max	*p*-value ^1^
Age (years)	6.1	5.8	3.8	0.1	17.9	8.0	5.9	6.3	0.1	17.8	0.022
Weight (kg)	22.0	20.8	12.6	1.3	120.0	30.8	22.8	21.0	1.9	101.0	0.006
Duration of stay (days)	9.9	13.0	5.0	1.0	83.0	5.6	6.1	3.5	1.0	74.0	0.005

MPP: Medication preparation process; Man.: Manipulation; %: percentage; ^1^ Mann-Whitney U test independent samples.

**Table 3 pharmaceutics-12-00583-t003:** Therapeutic subgroup of manipulated active substances classified by ATC code [25].

ATC Code	Therapeutic Subgroup	n	%	Active Substances (n)
N03	Antiepileptics	64	27	potassium bromide (4), phenobarbital (18), topiramat (10), lamotrigine (8), vigabatrin (6), valproic acid (4), oxcarbazepine (5), lacosamide (2), brivaracetam (3), clonazepam (2), gabapentin (2)
A11	Vitamins	47	19.8	cholecalciferol (46), cholecalciferol/sodium fluoride (1)
A02	Drugs for acid-related disorders	30	12.7	omeprazole (27), esomeprazole (2), sodium bicarbonate (1)
H02	Corticosteroids for systemic use	14	5.9	prednisolone (7), dexamethasone (4), hydrocortisone (3)
A12	Mineral supplements	8	3.4	potassium/citrate (4), sodium chloride (1), potassium chloride (2), zinc (2)
C02	Antihypertensives	8	3.4	clonidine (5), bosentan (1), sildenafil (2)
M03	Muscle relaxants	6	2.5	baclofen (6)
B03	Anti-anaemic preparations	5	2.1	ferrous gluconate (5)
H03	Thyroid therapy	5	2.1	levothyroxine sodium (4), thiamazole (1)
L04	Immunosuppressants	4	1.7	tacrolimus (1), everolimus (1), azathioprine (2)
N05	Psycholeptics	4	1.7	melatonin (1), clobazam (2), nitrazepam (1)
C07	Beta blocking agents	3	1.3	propranolol (1), metoprolol (1), atenolol (1)
C09	Agents acting on the renin-angiotensin system	3	1.3	enalapril (3)
G04	Urologicals	3	1.3	propiverine (2), trospium (1)
J01	Antibacterials for systemic use	3	1.3	sulfamethoxazole and trimethoprim (1), phenoxymethylpenicillin (1), nitrofurantoin (1)
M01	Anti-inflammatory and anti-rheumatic products	3	1.3	ibuprofen (3)
N07	Other nervous system drugs	3	1.3	pyridostigmine (3)
V03	All other therapeutic products	3	1.3	polystyrene sulfonate (1), calcium folinate (2)
	Other	21	8.9	examples: ondansetron (1), metformin (1)
	Overall	237	100	

ATC code: Anatomical Therapeutic Chemical classification code; %: percentage.

**Table 4 pharmaceutics-12-00583-t004:** Numbers of type of manipulation observed in this study.

Type of Manipulation	n	%
Tablet split/cut/broken	80	33.8
**Tablet split/cut/broken + other manipulation(s)**		
Tablet split/cut/broken, then dissolved/suspended/dispersed/diluted in liquid	35	14.8
Tablet split/cut/broken, then dissolved/suspended/dispersed/diluted in liquid, then a proportion of the liquid given	7	3.0
Solid dosage form dissolved/suspended/dispersed	59	24.9
**Solid dosage form dissolved/suspended/dispersed + other manipulation(s)**		
Tablet dissolved/suspended/dispersed/diluted in liquid, then a proportion of the liquid given	25	10.5
Capsule opened, then content dissolved/suspended/dispersed, then a proportion of the liquid given	3	1.3
Powder for intravenous infusion dissolved in liquid, then a proportion of the liquid given	3	1.3
**Other**		
Tablet crushed/mortared	2	0.8
Tablet crushed/mortared + other manipulation(s)	2	0.8
Liquid formulation diluted in larger volume	6	2.5
Withdrew a defined volume/dose from a container (ampoule) of liquids for intravenous use	3	1.3
Withdrew a defined volume/dose of powder for oral suspension from a sachet	2	0.8
Counted minitablets	2	0.8
Counted minitablets, then suspended the defined number of minitablets in liquid	1	0.4
Other	7	3.0
**Overall**	**237**	**100.0**

%: percentage.

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
