# Peer review of "Manipulation of Medicinal Products for Oral Administration to Paediatric Patients at a German University Hospital: An Observational Study"

_pharmaceutics, 2020, doi:10.3390/pharmaceutics12060583_

Round 1

Reviewer 1 Report

I have read with interest the article entitled "Manipulation of Medicinal Products for Oral Administration to Paediatric Patients at a German University Hospital: an Observational Study". This represents an important study, addressing a vital aspect of paediatric drug therapy, which highlights the ongoing concern for patients and healthcare professionals with regards to availability of age-appropriate medicines for children.

The manuscript is well written and logically structured, whilst the results are clearly presented and appropriately discussed.

There are some question marks over certain elements (as annotated on the edited pdf file attached), but these do not detract from the overall quality of the study and I would highly recommend that this be published, following minor revisions.

Author Response

We thank the reviewer for this positive evaluation of our manuscript and the constructive and valuable input. Please find the response to your comments below.

Point 1 (line 18): It would be best to maintain consistency and state numbers.

Response 1: It is a journal requirement to write out the numbers at the beginning of a sentence. This information is retrieved from the guidelines for authors of MDPI "Numbers at the beginning of a sentence should be written in full, i.e. 152 mL must be written as: One hundred and fifty two milliliters." (https://www.mdpi.com/authors/english-editing).

Point 2 (line 23): As above, state 56% to maintain consistency.

Response 2: Please see response to Point 1.

Point 3 (line 54): Is this necessary?

Response 3: You are right; this is not relevant. We deleted the sub-clause "including one systematic review from 2017". The sentence now reads as follows: "Several publications revealed that dose accuracy and weight uniformity highly depend on the manipulation method used [19-21]." (line 53-54).

Point 4 (line 62): for

Response 4: The expression "to express" has been changed to "for". The sentence now reads as follows: “However, the data do not allow for a general statement that can be applied to other countries.” (line 601-61).

Point 5 (line 66): a German hospital? I think that it is important to clarify the setting here.

Response 5: Thank you for this comment. We changed the wording to "paediatric wards in a German university hospital". The sentence now reads as follows: “The aim of this study was, therefore, to investigate the nature, frequency and prevalences of manipulations of medicinal products before oral administration on paediatric wards in a German university hospital by direct observation and to determine preventability of these manipulations.” (line 63-66).

Point 6 (line 94): Who was the observer(s) and were they suitably qualified to identify the risk? I've no doubt they were, but I think it would be worth clarifying here.

Response 6: In section 2.3, it is mentioned that a trained pharmacist carried out the observation (line 79 f.). Throughout the entire study period, only this person performed the observation. The training was carried out in the scope of a pilot study with a duration of approximately two weeks.

Point 7 (line 96): is not in line?

Response 7: Thank you very much for this comment. The "not" has been added in the file. The sentence is now as follows: "An example of a harmful situation is if the dispensed dose is not in line with the prescribed dose." (line 94)

Point 8 (line 98): I would suggest that 'dispensing' would be the better term.

Response 8: The term has been changed to 'dispensing'. The sentence now reads as follows: "MPP was defined as all processes related to the preparation and dispensing of the patient individually prescribed dose by the nursing staff before oral drug administration." (line 96).

Point 9 (line 108 f., Table 1): Can this really be classed as a manipulation? How else would you get the medicine out of the packaging (you don't class opening of bottles or popping of blisters as a manipulation)? Indeed, this isn't stated again in, for example, table 4.

Response 9: Thank you for this comment. You are right, opening of a sachet is mandatory to administer the content and is therefore not classified as manipulation. In this table, this bullet point was meant to be the first step of the manipulation. In the second step, only parts of the sachet were administered. The first point was deleted to avoid misunderstanding. Moreover, we adapted the following points in terms of consistency as follows:

"Dispersion of content in liquid and administration of a proportion.

Administration of a proportion of the content." (Table 2, line 106)

Point 10 (line 153): Please ensure that this and other tables are not split over two pages in the final version.

Response 10: We have ensured that no table is split over two pages in the final version.

Point 11 (line 153, table 2): I think that this is an interesting point that is not sufficiently addressed in the subsequent discussion - was the number of patients with off-label manipulation in this group (ward B) a consequence of the type of ward (and, therefore, the medicines that were generally administered)? Were there more patients with a tube (as it is a gastro ward)? All other data points are broadly similar, but this stands out to me.

Response 11: Thank you very much for pointing this out. While addressing this question, we noticed an error in the definition of the wards in the methods. Ward A was the one with focus on infectious and gastroenterological diseases, Ward B the one with the neuropaediatric focus. This fault has no consequences on the content of the manuscript, as the results were interpreted correctly. It was only necessary to correct the corresponding definition of ward A and B in the methods. It is now stated: "One ward had the focus on infectious and gastroenterological diseases (23 beds, Ward A) and the other one on neuropaediatrics and metabolic diseases (23 beds, Ward B)." (line 72 – 73).

Ward B was the ward with neuropaediatric focus. The higher prevalence of off-label manipulations was indeed a consequence of the patient population on this ward. On the one hand, there were many patients with developmental disabilities, which were, for example, not able to swallow solid dosage forms. On the other hand, there were many patients receiving antiepileptic drugs. These drugs need to be slowly titrated to the effective and save maintenance dose. Although some newer antiepileptic drugs are available in formulations with broad dose flexibility, there are still many drugs, which are not available in very small dose steps. This leads to the necessity of manipulations and explains the higher prevalence on this ward. This topic was already discussed in line 293 and the following. Besides, we included the following sentence regarding Ward B in line 299 – 301: "This does also explain why the ward with neuropaediatric focus showed a higher prevalence of off-label manipulations compared to the other ward with focus on infectious and gastroenterological diseases."

Point 12 (line 168, 171): As with the abstract, consistency in reporting numbers would be useful.

Response 12: Please see response to Point 1.

Point 13 (line 183 f.): Syntax changed slightly here, as the drug release was not analysed to determine if it had been affected by manipulation (even though there is a likelihood that it would be).

Response 13: Thank you for this reasonable change. The modified drug release itself was indeed not studied. We changed the sentence as recommended to "Manipulated medicinal products with modified drug release were Antra mups® (omeprazole, n = 21), Orfiril® long (valproic acid, n = 4), Beloc-zok® (metoprolol-succinate, n = 1) and Circadin® (melatonin, n = 1)." (line 181-183)

Point 14 (line 199): included

Response 14: We changed the fragment "were, i.a." to "included" as recommended. The sentence now reads as follows: "Other manipulation types included dilution of a liquid dosage form in a larger volume (2.5%), crushing of tablets (0.8%), and subsequent manipulation (0.8%) or counting of minitablets (0.8%) and subsequent suspension of the defined number of minitablets in liquid (0.4%)." (line 199-202)

Point 15 (line 213): Could you say 'In general, the younger the patient...', as there are some instances (from looking at the figure below) where this may not be the case (e.g. infants/toddlers have a lower proportion of >1-step manipulation than pre-school children for split and dispersed)? Evidently there is a correlation (a borne out by the stats), but this needs clarifying.

Response 15: You are right that there are instances where younger patients have a lower proportion of >1-step manipulations. The higher proportion of 1-step manipulation in infants and toddlers can be explained by the fact that children in this age group receive daily cholecalciferol 500 IU, which is obtained by splitting a 1000 IU tablet in the hospital setting. Thank you very much for informing us that this needs to be clarified. We added the term “in general” at the beginning of the sentence (line 215). The sentence now reads as follows: “In general, the younger the patient, the more manipulation steps were observed.” Moreover, we added the following clarification in the discussion (line 266-268): "The frequent use of cholecalciferol tablets can also be considered as a reason for the high proportion of one-step manipulations in infants and toddlers. Nevertheless, this study demonstrated that, overall, younger age is associated with an increased number of manipulation steps."

Point 16 (line 247): Was this noted (even if informally) during the observations? Also, was this difference statistically significant?

Response 16: This was noted informally during the observation when the observer talked with the nurses about the patients and their medicines. There was no statistical significance between the frequencies of manipulations in preschool-children compared to other age groups (line 159-162).

Point 17 (line 280): Expand this acronym.

Response 17: We expanded this acronym. The sentence now reads as follows: “Among the preventable manipulations, three cases could have been avoided if a medicinal product licensed through a Paediatric Use Marketing Authorization (PUMA) would have been used (Slenyto® (melatonin), Hemangiol® (propranolol) and Alkindi® (hydrocortisone)).” (line 283).

Point 18 (line 286): whom it is not of economic interest

Response 18: We adapted the sentence as recommended to "The PUMA needs to become more attractive to pharmaceutical companies, for whom it is not of economic interest to develop a paediatric formulation of a generic drug." (line 290-292)

Point 19 (line 299): This is a very important point - did you consider looking at this detail within your results, to determine the proportion of NTIs that were manipulated?

Response 19: Thank you for this comment. This is, indeed, a very interesting point. However, we did not look in this in detail. The main challenge is to define when an active substance classifies as NTI. We added the following sentence to address this issue (line 305 f.): "In this evaluation, about one-third of manipulations affected narrow therapeutic index drugs (NTIs), if the therapeutic subgroups of antiepileptics, immunosuppressants and antihypertensives were considered as NTIs.”

Point 20 (line 328): Moreso, it is vital to consider the formulation type (i.e. how the modified release is imparted on the dosage form) in order to determine the potential impacts of manipulation. It would be worth a comment in this regard here.

Response 20: Thank you for this relevant comment. We added the following sentence to make this important information available in the paper: "In this context, it is also vital to consider the mechanism of drug release in order to determine the potential impact of manipulation of a modified release formulation." (line 341-343).

Reviewer 2 Report

Understanding the nature, frequency, and preventability of oral drug manipulation is an exciting research topic. The manuscript by J. Zahn et al. describes a series of experiments that address the prevalence, frequency, substance group, age group, and dosage form of manipulation. The observational study provides scientific guidance for pharmaceutical companies to develop more age-appropriate formulations and to prevent off-label manipulation. I am in favor of publication after minor revisions. The abbreviation PUMA was not explained.

Author Response

We thank this reviewer for positive feedback. We have added the abbreviation of PUMA (Paediatric Use Marketing Authorization) accordingly (line 284).

Reviewer 3 Report

The aim of this paper is the investigation of pharmaceutical forms manipulations of medicinal products performed in hospitals before oral drug administration to paediatric inpatients in Germany. The study evidenced the high manipulation of medicinal products on paediatric wards in Germany that requires appropriate guidelines.

Appropriate pharmacotherapy for paediatric population is a very important topic that requires more in deep investigation and effort from pharmaceutical technology research and industries. I found the paper interesting and well written, just few corrections and adjustments can improve the paper clearness.

  1. A list of abbreviations could be useful.
  2. The Authors should more in deep explain which kind of informations should  be reported in the SmPC
  3. Page 2 line 61 Can the Author report the studies duration and the number of cohorts of the commented paper? Days? Weeks?
  4. Galenic preparations have not been taken into consideration. Can the Authors comment the advantage of disadvantage of this solution (also from an economic point of view).
  5. Table 2 Why the sum of female and male patients does not match the total? 127+149= 276 and not 193
  6. It’s a pity that, during 5 months, clinical outcomes and adverse events hasn’t been registered. Is not possible to trace the data from the medical records? It would be very interesting.

Author Response

We thank the reviewer for constructive and valuable feedback, which helped to improve the clearness of this paper. Please find our responses to the comments below.

Point 1: A list of abbreviations could be useful.

Response 1: A list of abbreviations was added at the end of the manuscript (line 427 f.).

Point 2: The Authors should more in deep explain which kind of informations should be reported in the SmPC

Response 2: We thank the author for this important point. We advocate that information on feasible manipulations should generally be made available in the SmPC. To make this clear we added the following sentence: "We, therefore, advocate that information on feasible manipulations should generally be made available in the SmPC. This information should address the divisibility of tablets, especially whether a tablet may be divided into equal doses, mortarability as well as the possibility to disperse a tablet in liquid. Furthermore, manufacturers are encouraged to address the option to administer a medicinal product via enteral feeding tube and provide guidance, e.g. solvent or Charrière size of the tube." (line 353 f.)

Point 3: Page 2 line 61 Can the Author report the studies duration and the number of cohorts of the commented paper? Days? Weeks?

Response 3: The reviewer refers to the section where we justified the need for our study, as data of other studies do not allow expressing a general statement that can be applied to other countries.

The duration of observation of the three referenced studies was between one and two weeks per ward. Bjerknes et al. (2017) observed for eight weeks in total, van der Vossen et al. (2019) for six weeks in total and the total study duration of Richey et al. (2013) remains unknown. Collected data regarding the duration of the study and the number of cohorts are given in the table below.

The number of patients is not given for Richey et al. (2013) and Bjerknes et al. (2017). Van der Vossen et al. (2019) observed the administrations of 35 patients.

We felt that the duration of observation of 1 – 2 weeks per ward is not sufficient to determine the real frequency of manipulations. Therefore, we decided to directly observe the nursing staff at one ward for a longer period. We saw fluctuations of manipulations per week depending on the work-flow of the individual staff and the patient collective on the ward.

What else can be added is that the study by Richey at al. (2013) was not designed to determine the real frequency of manipulations as they identified manipulations prospectively via daily prescription reviews and not by direct observation.

We adapted the introduction by adding the duration of the studies by Bjerknes et al. and van der Vossen et al. (line 57 and 60). Moreover, we deleted the sub-clause "due to the short duration of these studies and the relatively small number of cohorts" (line 61).

The passage now reads as follows: "A study from Norway, which was conducted for eight weeks in two hospitals, showed that 17% of 3070 orally administered drugs to paediatric patients were manipulated not according to the relevant SmPC [9]. Recently, van der Vossen et al. [22] revealed that 37% of oral administrations to paediatric inpatients in the Netherlands were manipulated in six weeks of observation. However, the data do not allow for a general statement that can be applied to other countries." (line 57).

Study

Duration of study (observation)

Number of cohorts

Richey et al. 2013

Observations were conducted in blocks of two weeks (three sites, 21 different inpatient areas).

Number of drug administrations: not given

Number of patients: not given

Number of manipulations: 310

Bjerknes et al. 2017

The information was collected for one-four-week period for each hospital (two hospitals à 8 weeks in total).

Number of drug administrations: 3070

Number of patients: not given

Number of manipulations: 509

van der Vossen et al. 2019

Observation took place for one week in each of the six wards (6 weeks in total)

Number of drug administrations: 115

Number of patients: 35

Number of manipulations: 42

Point 5: Galenic preparations have not been taken into consideration. Can the Authors comment the advantage of disadvantage of this solution (also from an economic point of view).

Response 5: There were different reasons for not considering galenic preparations (extemporaneous formulations). One was that, in Germany, every hospital or public pharmacy can manufacture extemporaneous formulations. Therefore, the quality of the product can be questioned as processes may not be well standardised as in pharmaceutical companies. In addition, the production is time-consuming for the (hospital) pharmacies, the quality and shelf life is often worse than that of a ready-to-use licensed medicinal product. Most importantly, quality, safety and efficacy of extemporaneous formulations are usually not studied in patients and, therefore, they are off-label.

By not taking into account extemporaneous formulations when rating the preventability of manipulations, the numbers of "non-preventable manipulations" are higher. However, pharmaceutical companies should be encouraged to develop age-appropriate formulations. The overall aim is to have available a broad spectrum of age-appropriate, ready-to-use medicinal products in the future.

Point 6: Table 2 Why the sum of female and male patients does not match the total? 127+149= 276 and not 193

Response 6: Thank you very much for pointing this out. We are sorry, this was a transmission error. We have corrected the valus in Table 2 accordingly (line 151 f.).

Point 7: It's a pity that, during 5 months, clinical outcomes and adverse events hasn't been registered. Is not possible to trace the data from the medical records? It would be very interesting.

Response 7: Indeed, it would have been very interesting to investigate the clinical outcomes or adverse events following administration of manipulated medicinal products. This was not done during the study and we believe that retrospective analysis will not be appropriate. However, we will certainly include this in our next investigation.